# Pronounced Inter-Individual Variation in Plasma Cortisol Response to Fluoxetine Hydrochloride in the Pig

**DOI:** 10.3390/ani10030504

**Published:** 2020-03-18

**Authors:** Laura E. Marsh, Robyn Terry, Alexandra L. Whittaker, Stefan Hiendleder, Cameron R. Ralph

**Affiliations:** 1School of Animal and Veterinary Sciences, The University of Adelaide, Roseworthy Campus, Roseworthy, SA 5371, Australia; laura.latimer-marsh@adelaide.edu.au (L.E.M.); alexandra.whittaker@adelaide.edu.au (A.L.W.); 2South Australian Research and Development Institute, Livestock Sciences, Roseworthy Campus, Roseworthy, SA 5371, Australia; robyn.terry86@gmail.com (R.T.); drcameronralph@gmail.com (C.R.R.); 3Davies Research Centre, School of Animal and Veterinary Sciences, the University of Adelaide, Roseworthy Campus, Roseworthy, SA 5371, Australia; 4Robinson Research Institute, the University of Adelaide, North Adelaide, SA 5006, Australia

**Keywords:** cortisol, antidepressant, fluoxetine hydrochloride, pig, affective state, welfare

## Abstract

**Simple Summary:**

Holistic animal welfare assessment requires measures for emotional (affective) state, in particular positive states. Pharmacological agents such as antidepressants that create a predictable positive affective state can be valuable tools to assess novel welfare biomarkers. However, efficacy of pharmacological action in the brain needs to be demonstrated before such an approach is applicable. Counterintuitively, in humans and sheep, effective delivery of antidepressant agent, i.e., selective serotonin reuptake inhibitors, has been demonstrated by an increase in downstream cortisol levels. Here, we tested the efficacy of measuring circulating cortisol as an indicator for effective delivery of a single intravenous dose of the selective serotonin reuptake inhibitor fluoxetine hydrochloride to the pig brain. Antidepressant treatment resulted in increased plasma cortisol levels 15–165 min after treatment as compared with saline controls, suggesting that, similar to the other species, plasma cortisol is an indicator of fluoxetine hydrochloride efficacy. However, individual cortisol profiles of pigs treated with the antidepressant were highly variable with either the expected—an unorthodox, or no response. We conclude that significant inter-individual variation in cortisol response currently precludes the use of cortisol as a reporter for fluoxetine hydrochloride efficacy in the pig. These data need to be verified in a larger study.

**Abstract:**

Animal welfare assessment requires measures for positive affective state. Pharmacological agents that manipulate affective state can be used to evaluate novel biomarkers for affective state assessment. However, to validate that an agent has modified brain function, a reliable indicator is required. Circulating cortisol has been used as a reporter for effective delivery of the antidepressant selective serotonin reuptake inhibitor (SSRI) fluoxetine hydrochloride to the brain in humans and sheep. Here, we tested cortisol as a reporter for effective delivery of fluoxetine hydrochloride to the pig brain. We hypothesized that following administration of SSRI, innervation of the serotonergic reward pathway would result in activation of the hypothalamic-pituitary-adrenal (HPA) axis, leading to increased circulating cortisol levels. Large White-Landrace gilts received either a single intravenous dose of 100 mg fluoxetine hydrochloride suspended in 10 mL saline (n = 4), or 10 mL saline alone (n = 4). Blood samples were collected every 15 min for one hour prior to, and six hours post-treatment. The interaction of treatment x time on mean plasma cortisol levels between 15–165 min post-treatment was significant (*p* = 0.048) with highest cortisol concentrations of SSRI treated pigs versus controls (+ 98%) at 135 min post-treatment. However, individual cortisol profiles after SSRI treatment revealed high inter-individual variation in response. We conclude that, while combined data imply that plasma cortisol may be a readout for SSRI efficacy, inter-individual variation in SSRI response may preclude application of this approach in the pig. Given the current limited sample size, further research to confirm these findings is needed.

## 1. Introduction

The assessment of animal affective state can be challenging, in particular the evaluation of positive states. At present, behavioral measures and affective bias tests are the predominant assessment methods for positive states [1]. However, these methods are less suited to a production environment, because they are time-consuming and arguably subjective [2,3]. There is, therefore, an urgent need to identify and validate novel physiological and molecular markers of positive affect, such as miRNA [4,5], to complement or even replace behavioral [6] measures.

Validation of novel biomarkers for affective state requires robust means to manipulate affective state in a consistent manner. Pharmacological agents, including antidepressant selective serotonin reuptake inhibitors (SSRI) are candidates for this approach [7,8]. However, in order to use SSRI in validation experiments of novel biomarkers, a measure that demonstrates the effective delivery of the SSRI to the brain is required. In humans and sheep, effective SSRI delivery to the brain has been associated with an increase in adrenocorticotropic hormone (ACTH), leading to a downstream increase in cortisol level [9,10,11,12]. Rodent studies have reported similar findings using serotonergic inhibitors and/or 5HT1a receptor antagonists [13,14,15]. To our knowledge, no such data are available for the pig.

Here we tested whether plasma cortisol is a reliable indicator of effective delivery of SSRI to the pig brain. We hypothesized that pharmacological stimulation of the serotonergic system with an intravenous dose of the SSRI fluoxetine hydrochloride, would activate the HPA axis resulting in increased plasma cortisol levels. This peripheral cortisol response would thus provide evidence for successful activation of the serotonergic system in the brain of the pig.

## 2. Materials and Methods

### 2.1. Animals and Housing

Animal procedures were approved by the University of Adelaide Animal Ethics Committee (S-2018-053) and conducted in accordance with the Australian Code for the Care and Use of Animals for Scientific Purposes (NHMRC, 2013), and the Animal Welfare Act, 1985 (SA). Eight Large White-Landrace females at 18 weeks of age (mean weight 85 kg, range of 72–92 kg) were sourced and housed at the Roseworthy Piggery, South Australia. Animals were kept in individual stalls (240 cm × 60 cm) and thus restricted in their movement, and within sight of other individuals, throughout the experiment. Water was available *ad libitum* and 4 kg standardized grower feed (Barastoc MP Pig 1300, Ridley’s, Adelaide, South Australia) was provided every morning. The study was conducted in December, the southern hemisphere summer.

### 2.2. Treatment Protocol

Pigs were habituated to individual stalls for seven days prior to study commencement. To aid in adjustment to human presence, pigs had human contact daily. On day 1 of the study, topical local anesthetic (Xylocaine, Provet, Adelaide, Australia) was applied to the ear vein and catheterization performed under manual restraint with a rope snare. Catheter tubing was secured to the neck of the animal using adhesive tape (Elastoplast, Zebravet, Adelaide, South Australia). Computer-generated randomization (Microsoft Excel 2016, Microsoft Corporation) was used to assign pigs into two groups of 4 animals each. On day 2 all animals received either intravenous (i.v.) 100 mg SSRI fluoxetine hydrochloride (Complimentary Compounds, Ballina, NSW, Australia) dissolved in 10 mL 0.9% saline (Zebravet, Adelaide, South Australia) or i.v. 10 mL 0.9% saline at 8:00 am. The dose was chosen based on previous studies, where an initial 10 mL bolus injection containing 70 mg of fluoxetine hydrochloride, followed by a continuous infusion of 98.5 µg/kg/d for eight days was effective at increasing ACTH and cortisol in pregnant sheep [10,11]. The higher dose was chosen because we aimed to test cortisol response after a single intravenous injection. Considering previously published data and standardization applied to mitigate factors known to affect cortisol response such as age and breed [16] sex [17], feed intake [18] and level of exercise [19], the use of 4 animals per treatment group was deemed sufficient for this study to minimize animal usage; a retrospective power calculation with the acquired data revealed a power of 71%.

### 2.3. Sampling and Cortisol RIA

Blood sampling started at 7:00 am, one hour before treatment at 8:00 am, with sampling performed every 15 min until six hours post-treatment. Each sample of 3 mL blood was collected into 5 mL Lithium-Heparin coated tubes (Vacuette, Greiner Labortechnik, Kremsmünster, Austria). Samples were immediately centrifuged at 1000× *g* for 10 min and plasma stored at −20 °C until further analysis. Animals that had received SSRI treatment could not re-enter the commercial herd and were euthanized with 1 mL/10 kg of pentobarbital sodium (Virbac Pty Limited, Milperra, NSW, Australia). Saline treated animals reentered the commercial herd. Plasma samples were assayed for cortisol in duplicate by RIA following the manufacturer’s instructions (ImmuChem CT cortisol kit, MP Biomedicals, Orangeburg, NY, USA). Sensitivity of the kit was 0.17 pg/dL and intra and inter-assay coefficients of variation <15% and <10%.

### 2.4. Statistical Analysis

Statistical analyses were conducted with SPSS, Version 25 (IBM, Armonk, NY, USA). A linear mixed model analysis with time as the repeated measure was used to analyze the data. Normality and homogeneity of the dataset were tested by examining the correlation between the residuals and predicted values. As the data were not normally distributed, cortisol values were log10-transformed for the final analysis. The effect of animal weight on cortisol levels was tested in the model and removed due to lack of significance (*p* > 0.10). Statistical significance level was *p* < 0.05.

## 3. Results

A significant treatment by time interaction was observed between 15 and 165 min after treatment (*p* = 0.048). The greatest increase in mean plasma cortisol concentration of SSRI treated pigs as compared with saline controls (+ 98%) was measured 135 min post-treatment (Figure 1A). However, individual cortisol response profiles of SSRI treated animals varied considerably (Figure 1B). While the elevated cortisol profiles of SSRI treated Animals 5 and 7 were consistent with an SSRI induced cortisol response, two other SSRI treated animals displayed unorthodox cortisol response profiles (Figure 1B). Animal 6 did not respond to SSRI treatment, and Animal 8 revealed an initial spike in cortisol at 120 min post-treatment that quickly returned to baseline levels (Figure 1B). Saline treatment of control animals had no effect on circulating cortisol levels (Figure 1B).

## 4. Discussion

Here we tested whether measurement of circulating cortisol levels could be used as a reliable indicator of effective delivery of SSRI to the pig brain.

We observed a substantial increase in mean plasma cortisol levels after SSRI treatment 15–165 min post-treatment as compared with saline controls. This is consistent with activation of brain regions involved in reward processing and in agreement with observations in human and sheep [9,12]. However, examination of individual cortisol response profiles after SSRI treatment revealed an unexpected degree of inter-individual variation in cortisol response. While two SSRI treated pigs displayed the expected cortisol response curve with an initial peak followed by a gradual decline over time, two other pigs had a very different and unorthodox cortisol response. One of the cortisol profiles indicated a lack of response while the other one indicated a short spike in cortisol followed by a sudden return to baseline. Considering the standardization of factors known to affect cortisol response (i.e., sex, time of day, time of feeding, level of exercise [16,17,18,19,20], and finding no significant effect for the co-variate body weight, this degree of inter-individual variation was unexpected and has, to our knowledge, not been described previously. It is noteworthy that, in humans, psychiatric research has revealed individual variability in patient response to SSRI treatment, where genetic variation, environmental exposure and gene-environment interactions likely influence treatment outcomes [21,22]. We therefore propose the following explanations for the observed differences in individual cortisol response profiles after SSRI treatment in the pig: (1) inherent differences in the pharmacological pathway of the drug, including differences in receptor number, structure or function, or (2) differences in HPA axis responsiveness to SSRI, or (3) a combination of these. Regardless of the causes for the observed variation in response, in order to understand the dynamic relationship between the neurobiology of the serotonergic system and its effect on HPA activity, and thus cortisol, it appears essential that inter-individual differences are taken into account.

Furthermore, our data caution against over-reliance on statistically significant results obtained from group means without due regard for the individual data that constitutes the finding. Further research is needed regarding plasma cortisol as a biomarker of SSRI delivery in the pig.

## Figures and Tables

**Figure 1 animals-10-00504-f001:**
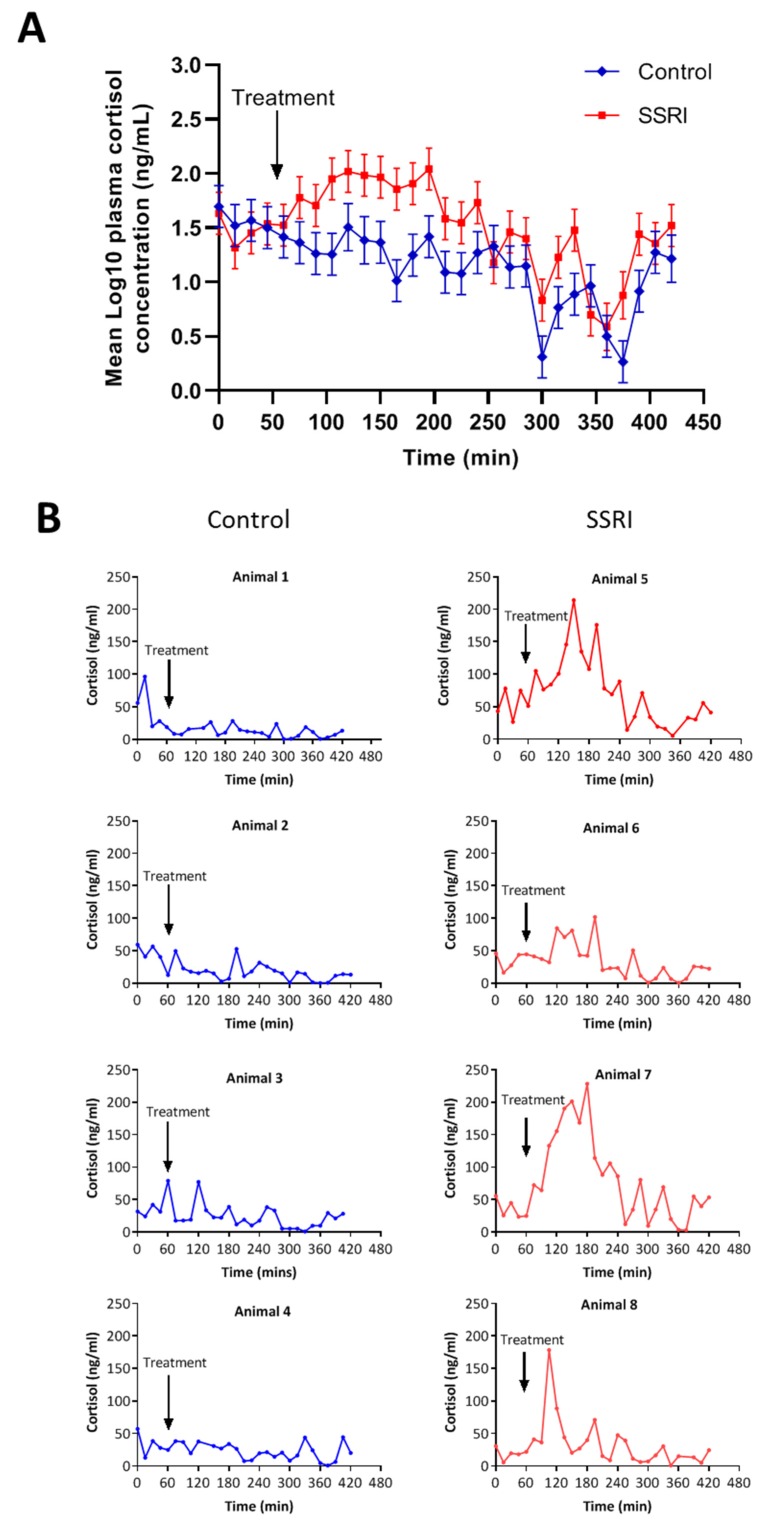
Plasma cortisol concentrations of pigs treated with either a single intravenous dose of the selective serotonin reuptake inhibitor (SSRI) fluoxetine hydrochloride (100 mg suspended in 10 mL saline) or saline control (10 mL). (**A**) Mean Log10 plasma cortisol concentration (ng/mL) ± SEM of animals with SSRI treatment in comparison to control animals that received saline. (**B**) Individual cortisol response profiles of control (animals 1–4) and SSRI treated (animals 5–8) individuals.

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
