# Peer review of "Pronounced Inter-Individual Variation in Plasma Cortisol Response to Fluoxetine Hydrochloride in the Pig"

_animals, 2020, doi:10.3390/ani10030504_

Round 1
Reviewer 1 Report
I have no further comments.
Reviewer 2 Report
Authors addressed all major comments received and the overall quality of the manuscript has improved. The paper is suitable for publication in its current form.
Reviewer 3 Report
The manuscript has been sufficiently revised and appears suitable for acceptance, now. This article provides information on the efficacy of measuring circulating cortisol as an indicator for effective delivery of fluoxetine hydrochloride to the pig brain. However, to validate an agent (in this article selective serotonin uptake inhibitor (SSRI) fluoxetine hydrochloride) modifying brain function, a reliable marker (e.g. Cortisol) is required.
Specific comment: Line 149: One bracket is missing
This manuscript is a resubmission of an earlier submission. The following is a list of the peer review reports and author responses from that submission.
Round 1
Reviewer 1 Report
Reviewer Comments
This article provides information on the efficacy of measuring circulating cortisol as an indicator for effective delivery of fluoxetine hydrochloride to the pig brain. However, to validate an agent (in this article selective serotonin uptake inhibitor (SSRI) fluoxetine hydrochloride) modifying brain function, a reliable marker (e.g. Cortisol) is required. Nowadays, affective states that refers to emotions or feelings play an important role in animal welfare but are difficult to assess. Therefore, novel biomarkers for assessment as cortisol release after administration of an SSRI to detect modification in brain function are of special scientific interest.
Broad comments:
It is difficult to understand (e.g. in Abstract and simple summary) the context of novel biomarkers and stress assessment due to the measurement of cortisol after fluoxetine hydrochloride injection. Please clarify this point.
The references are not conform to the guidelines of the journal. Headings sometimes are italic and not all authors are named
Two publications (10, 11) are cited that dose fluoxetine hydrochloride per weight (L86-88). As the mean weight of the gilts ranged from 72 to 92kg (+/- 8% of the mean body weight) and 70mg were administered in each gilt, please preclude or discuss the influence of different dosages (per kg) on the cortisol results in the treated animals
You mentioned a standardization according to previously published mitigate factors known to affect cortisol response (line 90ff) without citing these publications, which are in my opinion very relevant. Listing these considered, standardized factors would be important facts for the reader. Please provide this information for the reader to comprehend the study, the investigation schedule and environmental condition better.
As Pigs develop circadian rhythm of cortisol concentration about two weeks after birth, it would be important if you considered or standardized time of sampling in all animals and preclude the influence of daytime on cortisol concentration?
Specific comments
Lines 7, 9, 11, 13: numbering incorrect
Line 91: Citation of Reference incorrect
Line 139 please refer to standardized factors
Line 168, 169 & 171: Numbering incorrect
Reviewer 2 Report
The aim of the paper – validation of novel biomarkers for the affective state- is an admirable one.
To study the cognitive processes related to emotional states has great potential to advance and improve our understanding of animal welfare.
However, due to the very small sample size and the short duration of the experiment, I do not think this paper can be published as written. In my opinion, you recorded too little data to draw accurate conclusions
The introduction is poor and the objectives are too broad in their scope considering what is described later on in the manuscript. The citations are outdated (e.g. Fuller 1976, Bianchi 1994, it’s difficult to find on the web), or they are related to a chronic infusion (Morrison 2004)
Experimental design: Momentary assessments of cortisol depend on many several factors: between-individual differences, within-individual variation, and measurement error. Such complexities present well-recognized challenges to study design and interpretation of cortisol data. For this reason, it’s fundamental to collect multiple measurements from each individual for several days. In your study, the individual response profile lasts only 6 hours post-treatment. Furthermore, in figure 1 there are time points until 420 hours.
You mentioned the author “23” to describe your reasoning behind the sample sizes used, but I didn't find in the references
There are some mistakes in the references:
Line 168:
1. References
2. Whittaker, A.L. and L.E. Marsh, The role of behavioural assessment in determining ‘positive’ affective states
Reviewer 3 Report
The paper "brief report" is well written and explains adequately all obtained results. The only problem is in low amounts of the animals used for the evaluation of the effect of fluoxetine hydrochloride on cortisol level in pigs. The authors show the effects in individual animals and also the average values, which is correct. I agree with the conclusion which is written at the end of the paper: “Furthermore, our data caution against over-reliance on statistically significant results obtained from group means without due regard for the individual data that constitutes the finding. Further research is needed regarding plasma cortisol as a biomarker of SSRI delivery in the pig.” I recommend to stress also in the conclusion of abstract that the results were obtained only on small amounts of pigs (4) and thus further research is necessary for final conclusion.
Author Response
Please see attached comments